# Immunogenicity and Protective Efficacy of a Single Intranasal Dose Vectored Vaccine Based on Sendai Virus (Moscow Strain) against SARS-CoV-2 Variant of Concern

**DOI:** 10.3390/vaccines12070783

**Published:** 2024-07-16

**Authors:** Galina V. Kochneva, Gleb A. Kudrov, Sergei S. Zainutdinov, Irina S. Shulgina, Andrei V. Shipovalov, Anna V. Zaykovskaya, Mariya B. Borgoyakova, Ekaterina V. Starostina, Sergei A. Bodnev, Galina F. Sivolobova, Antonina A. Grazhdantseva, Daria I. Ivkina, Alexey M. Zadorozhny, Larisa I. Karpenko, Oleg V. P’yankov

**Affiliations:** Federal Budgetary Research Institution State Research Center of Virology and Biotechnology «Vector», Rospotrebnadzor, 630559 Koltsovo, Russia; g16kud@gmail.com (G.A.K.); zaynutdinov_ss@vector.nsc.ru (S.S.Z.); ira.shulgina.99@bk.ru (I.S.S.); shipovalov_av@vector.nsc.ru (A.V.S.); zaykovskaya_av@vector.nsc.ru (A.V.Z.); borgoyakova_mb@vector.nsc.ru (M.B.B.); star_ekaterina@rambler.ru (E.V.S.); bodnev@vector.nsc.ru (S.A.B.); sgf@vector.nsc.ru (G.F.S.); gaa@vector.nsc.ru (A.A.G.); ivkina_di@vector.nsc.ru (D.I.I.); lexazzador@mail.ru (A.M.Z.); lkarpenko1@yandex.ru (L.I.K.); pyankov_ov@vector.nsc.ru (O.V.P.)

**Keywords:** Sendai virus, Moscow strain, recombinant variant, vectored vaccine, SARS-CoV-2 variant of concern, S protein, immunogenicity, protection, single intranasal delivery

## Abstract

The mouse paramyxovirus Sendai, which is capable of limited replication in human bronchial epithelial cells without causing disease, is well suited for the development of vector-based intranasal vaccines against respiratory infections, including SARS-CoV-2. Using the Moscow strain of the Sendai virus, we developed a vaccine construct, Sen-Sdelta(M), which expresses the full-length spike (S) protein of the SARS-CoV-2 delta variant. A single intranasal delivery of Sen-Sdelta(M) to Syrian hamsters and BALB/c mice induced high titers of virus-neutralizing antibodies specific to the SARS-CoV-2 delta variant. A significant T-cell response, as determined by IFN-γ ELISpot and ICS methods, was also demonstrated in the mouse model. Mice and hamsters vaccinated with Sen-Sdelta(M) were well protected against SARS-CoV-2 challenge. The viral load in the lungs and nasal turbinates, measured by RT-qPCR and TCID_50_ assay, decreased dramatically in vaccinated groups. The most prominent effect was revealed in a highly sensitive hamster model, where no tissue samples contained detectable levels of infectious SARS-CoV-2. These results indicate that Sen-Sdelta(M) is a promising candidate as a single-dose intranasal vaccine against SARS-CoV-2, including variants of concern.

## 1. Introduction

The utilization of non-pathogenic or weakly pathogenic viruses for the generation of vaccine constructs is a promising direction in the prevention of human infectious diseases. Viral vector-based vaccines are safe and induce both innate and adaptive immune responses without the involvement of the entire dangerous pathogen. In addition, viral vectors have adjuvant properties through the expression of various pathogen-associated molecular patterns (PAMPs) and subsequent activation of innate immunity [1].

As the respiratory tract is the main point of entry for SARS-CoV-2, intranasal immunization is of great importance to halt this infection. Compared to injectable vaccines, intranasal droplet vaccines provide additional levels of protection such as antigen-specific S-IgA, effector CD8+ T-cells, and resident memory T-/B-cells through direct delivery of the antigen to the site of infection, the respiratory mucosa. The most important advantage of intranasal droplet vaccines is safety in use due to the avoidance of needles [2,3].

The Sendai virus (mouse parainfluenza virus type I, genus Respirovirus, family *Paramyxoviridae*) is well suited for these purposes, as it is a respiratory virus and is capable of limited replication in human bronchial epithelial cells [4], as well as in certain categories of dendritic cells [5], without causing disease. During this limited replication process, recombinant variants of the Sendai virus are capable of inducing a specific immune response to the foreign viral proteins they express within the human body [6]. The use of the Sendai virus as a vaccine vector also provides non-specific antiviral protection, as it is one of the most potent natural inducers of interferon. The Sendai virus was previously used to produce human leukocyte interferon before the era of recombinant proteins [7,8].

An important advantage of using the Sendai virus as a vaccine vector is the high stability of its genome. This stability is associated with an unusual property of the Sendai virus genome, as well as other paramyxoviruses, known as the “rule of six”. This rule denotes a strict polyhexameric genome length (6n + 0, where n is one nucleotide). This property contributes to a particularly low frequency of homologous recombination of paramyxovirus genomic RNAs [9]. Additionally, the replication of the Sendai virus occurs exclusively in the cytoplasm and not in the cell nucleus, which minimizes the risk of genetic integration of the viral genome into the host genome [10].

One potential problem in testing of Sendai virus-vectored vaccines in a laboratory rodent model is the susceptibility of rodents to infection by the virus. Among rodents, mice are the most susceptible to Sendai virus infection, but some mouse lines, such as C57BL/6J and BALB/c, are resistant and tolerate a high infectious dose (10^5^ EID_50_) of the virus [11,12,13]. It has also been shown that insertion of transgenes between the *P* and *M* genes in the Sendai virus genome results in significant attenuation of the virus [13]. In our study, we used BALB/c mice, and the transgene insertion into the vaccine construct was performed exactly between the *P* and *M* genes of the Sendai virus genome.

The vaccine properties of the Sendai virus have been actively studied in global practice. Thus, intranasal Sendai virus-based vaccines have successfully completed clinical trials against human parainfluenza type I and respiratory syncytial virus [14]. A number of recombinant Sendai viruses are undergoing preclinical studies as vaccines against human respiratory infections [15], including COVID-19 [10].

Despite the relative decline in the incidence of COVID-19, the need to develop vaccines, especially rapid response vaccines, remains due to the possibility of the emergence of variants of concern (VOCs) of SARS-CoV-2. In our work, we tested a prototype of a single-dose intranasal vaccine, which is an example of a rapid response vaccine against the delta and gamma VOCs of SARS-CoV-2.

According to currently available data, the most effective immunogen in the case of SARS-CoV-2 infection is a full-length copy of the spike S protein of the virus, in its native, non-optimized form [16].

Recently, there has been a debate on the safety of using S protein as an immunogen due to its possible involvement in the development of coagulopathy, which is one of the known complications of severe COVID-19. Studies that have examined the direct effect of spike proteins from SARS-CoV-2 variants on platelet activity and blood clotting have reported conflicting results. Thus, the work of Kuhn et al. showed that the SARS-CoV-2 S protein, through the RGD (Arg-Gly-Asp) motif, could weakly interact with some integrins on the surface of human platelets and trigger their stochastic activation. The authors suggest that such activation may be associated with the pathogenesis of COVID-19 and the occurrence of coagulopathies [17]. However, the work of Kusudo et al. showed that spike proteins from SARS-CoV-2 variants (alpha, beta, gamma, delta) had no effect on coagulation, activity, quantity, average volume platelets, and thromboelastography parameters in an ex vivo study [18].

To date, much evidence has accumulated on the efficacy and safety of COVID-19 vaccines, including vector vaccines that utilized the SARS-CoV-2 spike protein as an immunogen. A meta-analysis of eight randomized controlled trials of four COVID-19 vaccines based on mRNA, adenovirus, whole virion inactivation, and subunit vaccines, comprising 195,196 participants, found no statistically significant risk of thromboembolism and bleeding after vaccination compared with placebo [19]. A cumulative analysis of global data on the use of vaccines against COVID-19 (more than 300 million vaccinated individuals in Europe and USA) has shown that during infection with SARS-CoV-2 and the related disease (COVID-19), thrombosis occurs at least 100 times more often without vaccination than after vaccination [20]. The experience with the Russian Sputnik V vaccine (a vector vaccine based on recombinant adenoviruses expressing the SARS-CoV-2 spike protein) has shown that it prevents the severe course of COVID-19 and the development of fatal outcomes, pulmonary embolism, and venous and arterial thrombosis [21].

Based on the information provided, we used the full-length DNA sequence of the S gene from a SARS-CoV-2 natural isolate belonging to the B.1.617.2 lineage (delta VOC) as an immunogenic transgene in a recombinant variant of the Sendai virus.

In this paper, we present data on the engineering of a recombinant Sendai virus, Moscow strain, expressing the full-length spike (S) protein SARS-CoV-2 delta VOC, and the results of studying the immunogenic and protective properties of the recombinant virus in a single intranasal immunization of BALB/c mice and Syrian golden hamsters.

## 2. Materials and Methods

### 2.1. Cell Cultures

Rhesus macaque kidney cell culture LLC-MK_2_ (Flow Laboratories, London, UK) was used to titrate Sendai virus. Vero E6 African green monkey kidney cells (Cell Culture Collection of FBRI SRC VB “Vector”, Rospotrebnadzor) were used to titrate SARS-CoV-2 and determine the virus-neutralizing activity of animal blood sera. Recombinant 293-T7 cells stably expressing the T7 polymerase were kindly provided by Dr. Sergey V. Kulemzin (Institute of Molecular and Cellular Biology SB RAS, Novosibirsk, Russia).

All cells were maintained in DMEM nutrient medium with 4.5 g/L glucose (Invitrogen, Carlsbad, CA, USA), supplemented with GlutaMAX™ (Gibco, Miami, FL, USA), 10% FBS (HyClone, Logan, UT, USA), and antibiotic-antimycotic (Gibco, USA) at 37 °C and 5% CO_2_.

### 2.2. Chicken Embryos and Red Blood Cells of Animals

Chicken embryos of 11 days of age were obtained at Novo-Baryshevskaya Poultry Farm (Baryshevo village, Novosibirsk region, Russia).

The blood of a rooster and a guinea pig were obtained from the vivarium of FBRI SRC VB “Vector”, Rospotrebnadzor. Erythrocyte lavage and preparation of a working 1% suspension was carried out with 0.9% sodium chloride solution (saline solution).

### 2.3. Viruses

A recombinant variant of the Sendai virus expressing SARS-CoV-2 full S protein (Sen-Sdelta(M)) was generated based on the Moscow strain (GenBank: KP717417.1) [22] using a set of recombinant plasmid DNA described in [23]. A transgene corresponding to the full-size S protein of the SARS-CoV-2 delta VOC (Sdelta transgene) was obtained by RT-PCR using the primers presented in Table 1 and genomic RNA obtained from the SARS-CoV-2 HCoV-19/Russia/MOS-2406/2021 strain belonging to the lineage B.1.617.2 (delta). The HCoV-19/Russia/MOS-2406/2021 strain was isolated from an adult patient in June 2021, GISAD ID: [EPI_ISL_7338789].

The amplicon of the Sdelta transgene was inserted into the polylinker region of the genomic plasmid pSen2-MCS(M) using the restriction–ligation method at the BsiWI and BssHII restriction sites (Figure 1) to obtain the genomic plasmid DNA pSen2-CoVSpike2(M). A complete nucleotide sequences of pSen2-CoVSpike2(M) was verified by Sanger sequencing.

The recombinant Sen-Sdelta(M) was rescued as a result of transfecting 293-T7 cells with a set of four plasmid DNAs: a genomic plasmid (pSen2-CoVSpike2(M)) and three helper plasmids expressing the *N*, *P*, and *L* genes of the Sendai virus. Transfection was performed using Lipofectamine Plus Reagent (Invitrogen, USA) according to the manufacturer’s instructions. The 293-T7 cells together with the culture medium were frozen/thawed once 48 h post-transfection and incubation at 37 °C with 5% CO_2_. The transfection material was then transferred into the allantois cavity of 11-day-old chicken embryos (200 μL per egg), and the recombinant virus was isolated from the allantoic fluid after 72 h of incubation at 37 °C. The presence of the rescued virus was determined by a hemagglutination (HA) assay using 1% chicken red blood cell solution. For this purpose, 50 µL of allantois fluid from each egg was placed in a well of a round-bottom 96-well plate. Then, 50 μL of 1% chicken red blood cell solution was added to each well and the plate was incubated for 1 h at 4 °C. Allantois fluid from the eggs whose samples elicited a hemagglutination reaction was pooled and used to grow and purify recombinant virus.

The Sendai virus, both the original vector and recombinant, was grown in the allantoic fluid of 11-day-old chicken embryos for 72 h at 37 °C. The HA-positive samples were combined. The debris was pelleted by centrifugation at 1150× *g* for 10 min, and the supernatant was frozen at −40 °C. After thawing, the samples were centrifuged again at 12,000× *g* for 30 min at 6 °C, and the virus in the supernatant was pelleted by centrifugation at 100,000× *g* for 45 min at 6 °C. The supernatants were aspirated, and the pellets were resuspended in PBS, pH 7.4 (1/10 the initial volume), with the addition of magnesium chloride to a final concentration of 1 mM. The resulting viral suspension was sonicated three times using a cup horn sonicator. Each sonication was performed at 4 °C with ice for 1 min at 160 W, followed by vortexing for 30 s. The ice water was replaced between each sonication cycle. The product was then packaged and stored at −80 °C.

Titration of the Sendai virus was performed by plaque method on LLC-MK_2_ cell culture using guinea pig erythrocytes for virus visualization as described in [24]. The titer of the virus was expressed in the number of plaque-forming units (PFU) per 1 mL of suspension. The titer of the purified concentrated preparation of the Sendai virus was 1.5 × 10^9^ PFU/mL.

The following SARS-CoV-2 strains, including various variants of concern (VOCs), were used to evaluate the immunogenicity and protective efficacy of the vaccine: delta (HCoV-19/Russia/MOS-2406/2021, GISAID ID: [EPI_ISL_7338789]) and gamma (HCoV-19/Russia/SA-17620-080521/2021, GISAID ID: [EPI_ISL_6565014]). Additionally, the B.1.1 strain (hCoV-19/Russia/Omsk202118_1707/2020, GISAID ID: [EPI_ISL_1242008]), which shares homology with the Wuhan strain, was also included in the assessment. All strains were obtained from the State collection of causative agents of viral infections and rickettsioses FBRI SRC VB Vector, Rospotrebnadzor.

### 2.4. Western Blot Analysis

LLC-MK_2_ cells infected by recombinant or original vector strains of the Sendai virus were lysed in buffer: 50 mM Tris (pH 8.0), 5 mM EDTA, 150 mM NaCl, 0.1% SDS, 1 mM PMSF, and complete protease inhibitor cocktail (Roche Diagnostics, Mannheim, Germany). Samples were separated by 12% SDS–PAGE and transferred to a trans-blot nitrocellulose membrane (Bio-Rad, Hercules, CA, USA) by a wet blotting procedure (100 V, 500 mA, 90 min, 15 °C). The membrane was blocked with 5% dry milk (Fisher scientific, Waltham, MA, USA) in TBS for 1 h at RT on a shaker. Then, membrane was washed with TBS on a shaker 3 times (10 min at RT each time) and incubated with primary antibodies diluted in TBST containing 5% dry milk overnight at 5 °C on a shaker. To detect the spike protein of SARS-CoV-2, a human blood serum of COVID-19 convalescent (1:200) was used. The membranes were then washed with TBST on a shaker 3 times (10 min at RT each time) and incubated with goat anti-human-alkaline phosphatase-conjugated polyclonal IgG (1:3000) (Sigma-Aldrich, St. Louis, MO, USA) in TBST containing 5% dry milk for 1 h at RT. The secondary antibody was discarded and the membranes were washed with TBST on a shaker 3 times (10 min at RT each time). The immune complex was visualized by adding BCIP/NBT-Purple liquid substrate (Sigma-Aldrich, USA). The reaction was stopped by washing the membrane in distilled water. A GelDoc Go scanner (Bio-Rad, USA) was used for detection.

### 2.5. Laboratory Animals and Immunization Procedures

Six-week-old female golden Syrian hamsters (n = 16) and eight-week-old female BALB/c mice (n = 28) were obtained from the nursery of laboratory animals of the FBRI SRC VB “Vector”, Rospotrebnadzor. The hamsters and mice were housed in groups of 4 and 6–8 animals, respectively, under standard conditions, and had free access to food and water at all times. All manipulations with the animals, including immunization, infection, and tissue collection, were carried out after premedication with a combination of Zoletil 100 (Valdepharm, Val-de-Reuil, France) and Xyla (Interchemie, Harju maakond, Estonia).

The animals were divided into two equal groups—immunized and control. Immunization with the recombinant virus Sen-Sdelta(M) was carried out once intranasally at a dose of 10^5^ PFU per mouse and 10^6^ PFU per hamster. The volume of each inoculum was 100 μL for hamsters (50 μL in each nostril) and 10 μL for mice. The control groups received a similar amount of PBS, pH 7.4, in which the virus was diluted. Blood was collected from animals 28 days after vaccination, incubated for 1 h at 37 °C and 2 h at 4 °C, and then centrifuged at 7000× *g* for 10 min. The sera were deactivated by heating for 30 min at 56 °C and stored at −20 °C. The sera samples were used to detect total and virus-neutralizing antibodies to SARS-CoV-2. To assess the cellular immune response, 6 mice were used from the vaccinated and control groups, were euthanized on the 28th day after vaccination, and the spleens were collected and subjected to an analysis for the specific immune cell populations.

### 2.6. ELISA and Neutralization Assay for SARS-CoV-2

The determination of total IgG class antibodies to coronavirus was carried out using the commercial test system “SARS-CoV-2-ELISA-Vector” (FBRI SRC VB “Vector”, Rospotrebnadzor, Moscow, Russia). To detect IgG, HRP-conjugated goat anti-Syrian hamster IgG (Invitrogen, USA) or goat anti-mouse IgG (Biomedicals, Santa Ana, CA, USA) was used. The neutralizing activity of blood serum was determined on Vero E6 cell culture as described in [25] against three variants of SARS-CoV-2: hCoV-19/Russia/MOS-2406/2021 (delta VOC), HCoV-19/Russia/SA -17620-080521/2021 (gamma VOC), and Wuhan.

### 2.7. IFN-γ ELISpot and ICS

Splenocytes were isolated by pressing individual spleen through 70 and 40 μm cell filters (Jet BioFIL, Guangzhou, China). After removal of erythrocytes using a buffer for erythrocyte lysis (Sigma, USA), splenocytes were washed twice and resuspended in RPMI 1640 nutrient medium supplemented with gentamicin (50 μg/mL) and *L*-glutamine. Cell viability and concentration were determined by the trypan blue dye test (Bio-Rad, USA) on an automatic cell counter TC20 (Bio-Rad, USA). A pool of peptides from the SARS-CoV-2 S protein sequence restricted by MHC class I and II molecules of BALB/c mice at a concentration of 20 μg/mL each was used to stimulate cells (Table 2). Peptides were calculated using IEDB Analysis Resource tools and synthesized by AtaGenix Laboratories (Wuhan, China), the purity of peptides was >80%.

The intensity of T-cell immune response in immunized mice was determined by the number of IFN-γ-producing splenocytes using the IFN-γ ELISpot method. The analysis was performed using a Murine IFNγ ELISPOT Kit (with precoated plates) (Abcam, Waltham, MA, USA) according to the manufacturer’s instructions. Splenocytes were plated at 2.5 × 10^5^ cells per well and stimulated with a mixture of the peptides mentioned above. The cells were incubated for 18 h at 37 °C in the presence of 5% CO_2_. The number of IFN-γ-producing cells was counted using an ELISpot reader from Carl Zeiss (Germany).

Intracellular cytokine staining (ICS) was performed according to a standard protocol using flow cytometry. Splenocytes of 0.7 × 10^6^ cells were incubated with the peptide mixture (Table 2) or without it for 3 h at 37 °C and 5% CO_2_ followed by Brefeldin A (1 μg/mL) adding and 15 h incubation. Monoclonal antibodies produced by Biolegend (San Diego, CA, USA) against CD3 (clone 500A2), CD4 (clone GK1.5), and CD8 (clone 53-6.7) conjugated to A.F 700, BV 785, and FITC, respectively, were used for staining of surface markers. After fixation with 1% paraformaldehyde in PBS and permeabilization with 0.5% Tween 20 in PBS, the cells were stained to detect intracellular cytokines by anti-IFN-γ APC (clone XMG1.2) and anti-IL-2 BV 421 (clone JES6-5H4) (Biolegend, USA).

The samples were analyzed by using a ZE5 flow cytometer (Bio-Rad, USA) and Everest software 5.50.2100.

### 2.8. Virus Challenge

A SARS-CoV-2 challenge study was conducted in an animal biosafety level 3 (BSL-3) facility in compliance with the Russian Sanitary Rules and Regulations.

Thirty-five days after immunization, animals were infected by intranasal inoculation of 3.8 Log_10_TCID_50_ of SARS-CoV-2 gamma variant per mouse and 3.3 Log_10_TCID_50_ of delta variant per hamster. The animals were monitored daily after infection. The endpoint of the study on viral load in target organ tissues (nasal cavity and lungs) of mice and hamsters was set based on the timing of the peak of viral replication, according to our previous studies. After 5 days of infection for mice and 6 days of infection for hamsters, the animals were humanely euthanized with CO_2_. The lungs and nasal turbinates were collected, and 10% homogenates (*v*/*v* in PBS) were prepared to analyze the viral load.

### 2.9. Viral RNA Quantification

The viral load was assessed by RT-qPCR in the lungs and nasal turbinates as described in [26]. Briefly, the total RNA was isolated using the Riboprep kit (ILS, Russia), and the copy number of viral genomes was measured using a TaqMan real-time PCR reaction (qPCR) with the following primers:

5′-GTTGCAACTGAGGGAGCCTTG-3′ (forward),

5′-GAGAAGAGGCTTGACTGCCG-3′ (reverse), and

5′-FAM-TACACCAAAAGATCACATTGGCACCCG-BHQ1-3′ (probe).

The plasmid pJet1.2_SARS, containing a fragment of the SARS-CoV-2 genome (strain MN997409.1, nucleotide positions 28670-28826), was used as a reference for qPCR normalization. The number of copies of the viral genome was calculated using the DNA Copy Number and Dilution Calculator (Thermo Fisher).

### 2.10. Determination of Infectious Virus Titer in Tissue Homogenates

A 50% tissue culture infectious dose (TCID_50_) assay was used to quantify SARS-CoV-2 titers in lung and nasal turbinate homogenates. Vero E6 cells were seeded into 96-well plates and cultured for 24 h to form monolayers. Serial tenfold dilutions of 10% tissue homogenates were added to Vero E6 cell monolayers in eight replicates each. The plates were incubated for 4 days at 37 °C, and then the monolayers were stained with 0.2% Gentian violet solution. The presence of a specific cytopathic effect (CPE) was assessed visually by microscopic examination of the cell monolayer. Homogenate dilutions causing CPE in 50% of wells (endpoint dilution) were calculated according to the Reed–Muench method [27].

### 2.11. Statistics

Statistical data processing was performed using GraphPad Prism 9.0 software (Graph-Pad Software, Inc., San Diego, CA, USA). Quantitative data are provided as geometric mean titer or median with range and analyzed using nonparametric tests. Differences between groups (*p*-value) were obtained in pairwise comparison using the Mann–Whitney U criterion. A *p*-value less than 0.05 was considered statistically significant.

## 3. Results

### 3.1. Design and Characterization of the Recombinant Variant of Sen-Sdelta(M)

The recombinant variant Sen-Sdelta(M) of the Sendai virus (Moscow strain) was designed using a previously constructed set of four plasmid DNA, of which three express the *N*, *P*, and *L* genes of the Sendai virus (helper plasmids), and the fourth contains a full-size DNA copy of the genomic RNA of the Sendai virus, Moscow strain, which includes a structural element for the insertion and expression of transgenes between the *P* and *M* genes (pSen2-MCS(M)) (Figure 1a) [23]. The structural element includes a polylinker of five unique restriction sites, BsiWI, NruI, ClaI, AscI, and BssHII, followed by a termination signal for the transgene and an initiation signal for the subsequent *M* gene transcription by the Sendai virus RNA polymerase. The expression of the *N*, *P*, and *L* genes of the Sendai virus, as well as the synthesis of genomic RNAs of recombinant strains of the Sendai virus, is carried out from the corresponding plasmid DNAs under the control of T7 polymerase in 293-T7 cells. The accuracy of the size of the synthesized genomic RNA is controlled by ribozymes located at its 3’ and 5’ ends.

The Sdelta transgene represents a natural, unmodified DNA copy of the S gene from the SARS-CoV-2 hCoV-19/Russia/MOS-2406/2021 strain, which belongs to the delta variant (lineage B.1.617.2). The Sdelta transgene was obtained using the RT-PCR method and cloned as part of the Sendai virus genomic plasmid pSen2-MCS(M) (Figure 1a). The recombinant Sen-Sdelta(M) was rescued in 293-T7 cells. Isolation and development of the rescued virus were carried out in the allantoic cavity of 11-day-old chicken embryos.

The expression of the Sdelta transgene was evaluated by Western blot analysis in lysates of LLC-MK_2_ cells infected with the Sen-Sdelta(M) and in preparations of purified, concentrated Sen-Sdelta(M) virus from the allantoic fluid of chicken embryos. The results of the Western blot analysis are shown in Figure 1b, which shows that several forms of the S protein are detected in lysates of cells infected with Sen-Sdelta(M) (lane 3). Forms with molecular weights of approximately 190 kDa and a doublet at 90–115 kDa appear to correspond to the SARS-CoV-2 full-length S protein and the S1/S2 cleavage products, respectively [16,28]. The form with a molecular weight above 260 kDa corresponds to the trimeric form of the S protein, which is also detected in the positive control sample (lane 8), and represents the trimeric form of the recombinant SARS-CoV-2 delta variant S protein produced in transformed CHO-K1 cells [25].

The S protein is also detected in the preparations of purified Sen-Sdelta(M) virus obtained from the allantoic fluid of chicken embryos (lane 7). Such an effect has previously been shown for another recombinant paramyxovirus, Newcastle disease virus expressing the SARS-CoV-2 S protein transgene, and it is due to the incorporation of the S protein into the virions of the recombinant virus [29]. In our purified and concentrated preparations of the Sen-Sdelta(M) virus, the S protein is represented almost entirely by the processed to the S1/S2 cleaved form, as has been previously shown for purified SARS-CoV-2 preparations [30]. In the medium of LLC-MK_2_ cells infected with Sen-Sdelta(M), the processed S1/S2 form of the SARS-CoV-2 S protein is also detected (lane 5), which apparently gets there as part of the budding virions of the recombinant virus from the surface of the cells.

### 3.2. Evaluation of the Immunogenicity and Protective Efficacy of Sen-Sdelta(M) in BALB/c Mice

To analyze the immunogenicity of Sen-Sdelta(M), BALB/c mice were divided into two groups, one of which was intranasally immunized once with Sen-Sdelta(M) while the other, the control group, received PBS in the same way. ELISA results of the immune sera showed that Sen-Sdelta(M) was able to induce a specific humoral immune response with an IgG class geometric mean reciprocal titer of 22,286 (Figure 2a).

As shown in Figure 2b, antibodies capable of neutralizing SARS-CoV-2, including the variants Wuhan, delta, and gamma, were detected in the blood serum of all immunized mice. The geometric mean titer (GMT) of antibodies that neutralize the SARS-CoV-2 delta variant was 184 (95% CI [89; 378]). A comparable neutralizing activity was also demonstrated for gamma variant, with a GMT of 160 (95% CI [68; 286]). However, a decrease in neutralizing activity was observed for the Wuhan strain to GMT 121 (95% CI [76; 194]), which was 1.52-fold lower compared to the delta variant. No neutralizing activity was detected in the serum of control group mice.

The T-cell response in the group of Sen-Sdelta(M) vaccinated mice was evaluated using ELISpot-IFN-γ and intracellular cytokine staining (ICS) methods (Figure 3). As shown in Figure 3a, vaccination induces a T-cell immune response with an average of 126 spots per 10^6^ splenocytes. This is significantly different from the magnitude of the immune response in the control group (*p* < 0.01).

Analysis of the T-cell response using ICS (Figure 3b) showed that a significant antigen-specific increase in IFN-γ-producing T-helper cells was observed in the group of Sen-Sdelta(M) vaccinated mice compared to the control group. Cytotoxic T-lymphocytes with CD8+IFN-γ+ phenotype in the vaccinated group, although showing a tendency to increase, did not differ significantly from the control mice. The group that received the vaccine also showed high levels of IL-2-producing cells for both T-lymphocyte populations, indicating effective induction of T-cell immunity as a result of immunization with Sen-Sdelta(M).

The protective efficacy of Sen-Sdelta(M) was evaluated against HCoV-19/Russia/SA-17620-080521/2021(gamma VOC). The choice of gamma variant SARS-CoV-2 was based on the susceptibility of BALB/c mice to it. On day 35 after vaccination, intranasal infection of experimental (Sen-Sdelta(M), n = 8) and control (Control, n = 8) groups of mice with SARS-CoV-2 virus was performed. The animals were euthanized on day 5 after infection to achieve the maximum viral load in the lungs according to previous studies. Figure 4 shows the results of SARS-CoV-2 detection in lungs and nasal turbinates. Viral RNA copies were quantified using RT-qPCR with specific primers for the SARS-CoV-2 E gene. The titer of the infectiously active SARS-CoV-2 in tissue homogenates was determined by TCID_50_ assay on Vero E6 cells.

In the lung homogenates of vaccinated animals (Figure 4a), a significant 11-fold reduction in the concentration of SARS-CoV-2 viral RNA (vRNA) was observed compared to the control group (*p* < 0.001). The concentration of vRNA in the control group was 3.17 × 10^4^ copies/mL (95% CI [8.61 × 10^2^; 9.85 × 10^6^]), while in the group vaccinated with Sen-Sdelta(M), it was 2.74 × 10^3^ copies/mL (95% CI [8.85 × 10^2^; 9.80 × 10^3^]). The infectious titer of SARS-CoV-2 was ten-fold reduced in the vaccinated group compared to the control (*p* = 0.002). Additionally, titer values below the detection threshold (≤1 log_10_ TCID_50_/mL) were observed in 62.5% of the lung tissue samples in the vaccinated group.

Even more pronounced differences were observed in the analysis of nasal turbinate homogenates. In the vaccinated group, the vRNA concentration was 3.81 × 10^3^; copies/mL (95% CI [3.59 × 10^2^; 4.93 × 10^4^]), which is 233 times lower (*p* = 0.007) than in the control group (8.9 × 10^5^ copies/mL, 95% CI [6.37 × 10^3^; 1.05 × 10^7^]). Additionally, 87.5% of nasal tissue samples from Sen-Sdelta(M) mice group had infectious titers below the detection threshold (titer ≤ 1 log_10_ TCID_50_/mL).

### 3.3. Assessment of Immunogenicity and Protectivity of Sen-Sdelta(M) in a Golden Syrian Hamster Model

The experimental group of golden Syrian hamsters (n = 8) received a single intranasal immunization with Sen-Sdelta(M), while a similar volume of PBS was administered intranasally to the control group (also n = 8). Twenty-eight days after immunization, the serum levels of total IgG and virus-neutralizing antibodies against SARS-CoV-2 were measured in all animals (Figure 5).

ELISA results, shown in Figure 5a, demonstrate that all vaccinated Sen-Sdelta(M) hamsters have a significant induction of a humoral immune response to the antigen of the delta variant of SARS-CoV-2, and the GMT of IgG class antibodies was 47,568 (95% CI [27,404; 82,570]).

High neutralizing activity of the blood sera of vaccinated hamsters against the delta variant of SARS-CoV-2 was also revealed with GMT 453 (95% CI [160; 1280]), which significantly differed from the control group (*p* = 0.03). All samples from the control group had titer values below the detection threshold. Predictably, lower neutralizing activity was detected relative to the Wuhan strain, with GMT values in the vaccinated group of 190 (95% CI [40: 640]).

On day 35 after vaccination, hamsters in the Sen-Sdelta(M) and control groups (n = 8 in each) were infected intranasally with the SARS-CoV-2 delta VOC. The endpoint of the study of viral load in the target tissues (nasal turbinates and lungs) was set on day 6, considering the peak of virus replication according to previous studies.

The study of viral load in the lungs (Figure 6a) showed that in the vaccinated group, only one out of eight animals had a detectable vRNA value, and the remaining samples were below the detection threshold (>36 Ct or <2.0 × 10^3^ copies/mL). The difference in median vRNA values between the vaccinated and control groups was 8.97 log_10_ (or 9.3 × 10^9^ times), with a high degree of statistical significance (*p* < 0.001).

SARS-CoV-2 RNA was detected in all nasal turbinate samples from vaccinated animals (Figure 6b), but at significantly lower concentrations compared to the control group (*p* < 0.001). Specifically, the concentration of vRNA in nasal turbinates of the vaccinated hamsters (6.11 × 10^4^ copies/mL; 95% CI [3.66 × 10^4^; 1.21 × 10^6^) was 4.39 log_10_ lower (or 2.47 × 10^4^ times) compared to the control group (1.51 × 10^9^ copies/mL; 95% CI [2.15 × 10^8^; 3.59 × 10^11^]).

All lung samples and nasal turbinates from the vaccinated hamsters did not contain detectable level of infectious SARS-CoV-2, while in the control group the infectious titer was 4 log_10_ TCID_50_/mL (Figure 6a,b) (*p* < 0.001).

## 4. Discussion

The SARS-CoV-2 variants of concern (VOCs) pose the greatest risk to public health due to their increased transmissibility, more severe illness (resulting in higher hospitalization or mortality rates), reduced effectiveness of antibodies created from previous infection or vaccination, and lower efficacy of treatments. The World Health Organization (WHO) has currently identified five VOCs: alpha, beta, gamma, delta, and omicron. These VOCs have the ability to evade the protective effects of most registered COVID-19 vaccines. Therefore, the WHO encourages the development and initiation of clinical trials for variant-specific candidate vaccines targeting these designated VOCs [31,32]. There is particular emphasis on vaccines that can generate mucosal immunity, as this is considered a critical area to address in the development of the next generation of COVID-19 vaccines [3].

Here, we report the construction and characterization of a mucosal vaccine based on a recombinant Sendai virus, Sen-Sdelta(M), which encodes the transgene of full-length SARS-CoV-2 spike (S) protein of the delta VOC. We have shown that the S protein of the SARS-CoV-2 delta VOC was expressed as a transgene in cells infected with the Sen-Sdelta(M) and was also incorporated into the purified recombinant Sendai virus particles obtained from the allantois fluid of embryonated chicken eggs. The phenomenon of incorporating recombinant SARS-CoV-2 spike protein into recombinant virus particles had been demonstrated earlier using another paramyxovirus, the Newcastle disease virus [29]. In both cases, recombinant S protein is presented predominantly as an S1/S2 cleaved form. This is due to the fact that the allantois fluid of chicken embryos contains a large number of active proteolytic enzymes involved in the processes of digestion, morphogenesis, and hemostasis of embryos [33]. Although SARS-CoV-2 predominantly utilizes the cellular protease Furin for S1/S2 processing [34], other proteases have been shown to be capable of such processing [35]. Apparently, one or more proteases contained in the allantois fluid of chicken embryos also have the ability to efficiently cleave the S protein of SARS-CoV-2.

To evaluate the immunogenicity and protectiveness of the Sen-Sdelta(M) vaccine construct against SARS-CoV-2 infection, we used two types of laboratory animals: golden Syrian hamsters, highly sensitive to coronavirus infection, and BALB/c mice, selectively sensitive to some strains of SARS-CoV-2. In both cases, we used a single intranasal delivery of Sen-Sdelta(M), as this route of vaccination is the best in terms of ease of use, as well as the least traumatic and safe compared to parenteral administration.

In the BALB/c mouse model, we have shown that a single intranasal immunization with the Sen-Sdelta(M) effectively induces a humoral response with the formation of neutralizing IgG antibodies to delta and gamma VOCs of SARS-CoV-2, as well as, although to a lesser extent, to the analog of the original Wuhan strain. Thus, the Sen-Sdelta(M) vaccine has sufficiently broad cross-reactivity against different SARS-CoV-2 variants and could possibly, even without any further modifications, be used for vaccination against newly emerging virus variants after appropriate validation.

T-cell response was assessed using the ELISpot-IFN-γ and ICS methods. ELISpot-IFN-γ analysis showed that the number of IFN-γ-producing splenocytes in Sen-Sdelta(M) immunized mice was significantly higher than that in a control group. ICS analysis revealed that both CD4+ and CD8+ lymphocytes producing IFN-γ and IL-2 in response to stimulation with peptide fragments of the SARS-CoV-2 spike (S) protein were generated in the Sen-Sdelta(M) vaccinated mouse group. The data obtained indicate the formation of a pronounced virus-specific T-cell response.

Mice vaccinated with Sen-Sdelta(M) were protected well against the challenge of the SARS-CoV-2 gamma VOC, as evidenced by an 11-fold and more than 200-fold decrease in the concentration of vRNA SARS-CoV-2 in the lungs and nasal turbinates, respectively, compared with the control. No detectable infectious virus was revealed in more than half of the lung (62%) and nasal turbinate (85%) samples, which also indicates that SARS-CoV-2 infection had been stopped in the mice.

When the protectiveness of the Sen-Sdelta(M) was evaluated in the golden Syrian hamster model (Figure 6), it was found that 100% of lung and nasal turbinate samples from vaccinated hamsters did not contain detectable level of infectious SARS-CoV-2 at the peak of infection after delta VOC challenge. The significant decrease in median infectious titer in the Sen-Sdelta(M) group compared to controls was 4 log_10_ in both organs (*p* < 0.001). These data are consistent with the findings of Ilinykh et al., who used a different paramyxovirus, human parainfluenza virus type 3, to generate a single-dose intranasal vaccine construct expressing the full-spike (S) protein of SARS-CoV-2 [16].

As follows from the presented data, effectiveness of the single intranasal Sen-Sdelta(M) vaccine in reducing the risk of developing SARS-CoV-2 infection varies between mice and hamsters. The immunogenicity of Sen-Sdelta(M) for both animal species does not differ significantly: the level of specific IgG and virus-neutralizing antibodies in mice is only 2 and 2.5 times lower than the corresponding indices in hamsters. However, the protective effect in the hamster model is more pronounced than in mice, which is obviously due to differences in the pathogenesis of SARS-CoV-2 infection in these animal species. The delta variant of SARS-CoV-2 does not induce an infectious process in mice [36], so we used a gamma variant adapted to mice [37] for SARS-CoV-2 challenge, which allowed us to obtain a high viral load in nasal cavity and lung tissues at the peak of infection (day 5) and thus model the disease process. Syrian hamsters are highly susceptible to infection with SARS-CoV-2, without the need for prior adaptation, and develop severe pneumonia similar to COVID-19 patients [38]. In this case, for SARS-CoV-2 challenge, we used the delta variant, which is the target of the Sen-Sdelta(M) vaccine, which obviously contributed to a more effective protection of the animals. Despite the differences identified, both experimental animal models demonstrated statistically significant immunogenicity and protectiveness of the Sen-Sdelta(M) vaccine.

The data obtained in our study indicate that recombinant Sen-Sdelta(M) is a promising vaccine candidate with protective properties against SARS-CoV-2 variants of concern already at a single intranasal administration. We also believe that Sen-Sdelta(M) or novel recombinants derived from our Sendai virus vector platform can be used for booster intranasal vaccination following parenteral administration of currently approved COVID-19 vaccines to induce a balanced systemic and local mucosal immune response in the respiratory tract.

## Figures and Tables

**Figure 1 vaccines-12-00783-f001:**
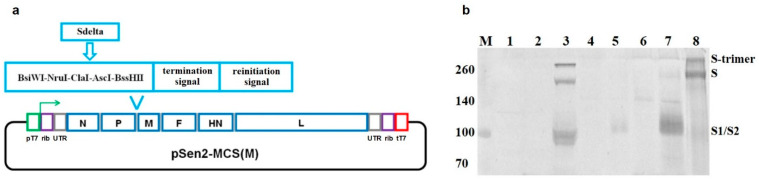
The schematic generation of recombinant virus Sen-Sdelta(M) (**a**) and Western blot analysis of Sdelta transgene expression (**b**). (**a**) The arrow indicates the direction of viral genomic RNA synthesis; pT7—promoter; TT7—terminator of T7 polymerase; rib—ribozyme. (**b**) M—molecular weight marker (kDa); 1—lysate of uninfected LLC-MK_2_ cells (negative control); 2—lysate of LLC-MK_2_ cells infected with the original vector strain of the Sendai virus; 3—lysate of LLC-MK_2_ cells infected with Sen-Sdelta(M); 4—cultural medium of LLC-MK_2_ cells infected with the original vector strain of the Sendai virus; 5—cultural medium of LLC-MK_2_ cells infected with Sen-Sdelta(M); 6—allantoic fluid of chicken embryos infected with the original vector strain of the Sendai virus; 7—allantoic fluid of chicken embryos infected with Sen-Sdelta(M); 8—S-trimer, developed in transformed CHO-K1 cells, 200 ng (positive control).

**Figure 2 vaccines-12-00783-f002:**
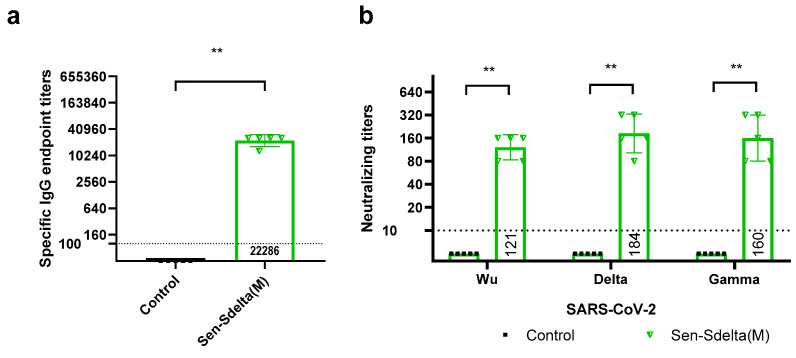
The specific IgG endpoint titers against delta VOC measured by ELISA (**a**), and the neutralizing titers against delta and gamma VOCs and the Wuhan strain of SARS-CoV-2 determined through neutralization assays on Vero E6 cells (**b**), in the blood serum of BALB/c mice at 28 days following intranasal immunization with Sen-Sdelta(M). Each data point represents an individual titer, with the tops of the histograms indicating the geometric mean titer (GMT), and vertical lines representing the 95% confidence intervals (95% CI). Horizontal drop lines denote the thresholds for IgG (<1:100) and neutralizing titers (<1:10), with values below these thresholds estimated as 1:40 and 1:5, respectively, for GMT calculations. ** *p* < 0.01.

**Figure 3 vaccines-12-00783-f003:**
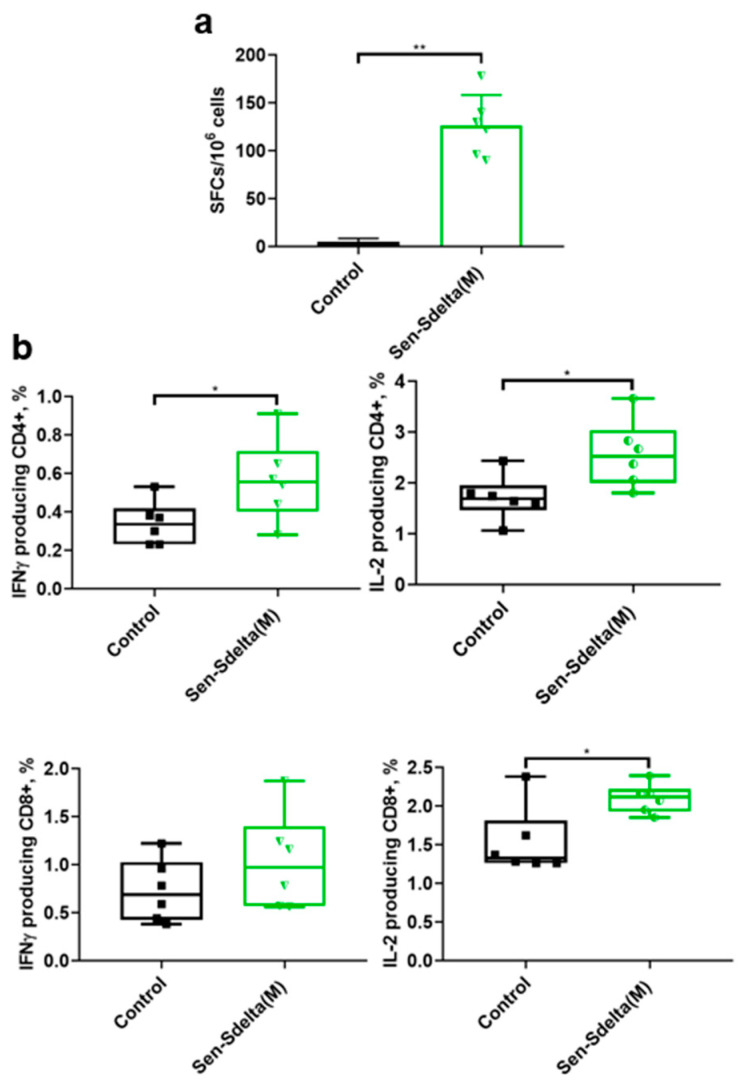
Cellular immune response in BALB/c mice evaluated by ELISpot (**a**) and ICS (**b**). (**a**) Number of splenocytes producing IFN-γ in response to specific stimulation, per 10^6^ cells. (**b**) Percentage of SARS-CoV-2-specific cytokine-producing CD4+ and CD8+ T-cells. Individual values are represented by dots, mean values by lines. The upper and lower edges of rectangles represent the boundary of the 95% CI. * *p* < 0.05, ** *p* < 0.01.

**Figure 4 vaccines-12-00783-f004:**
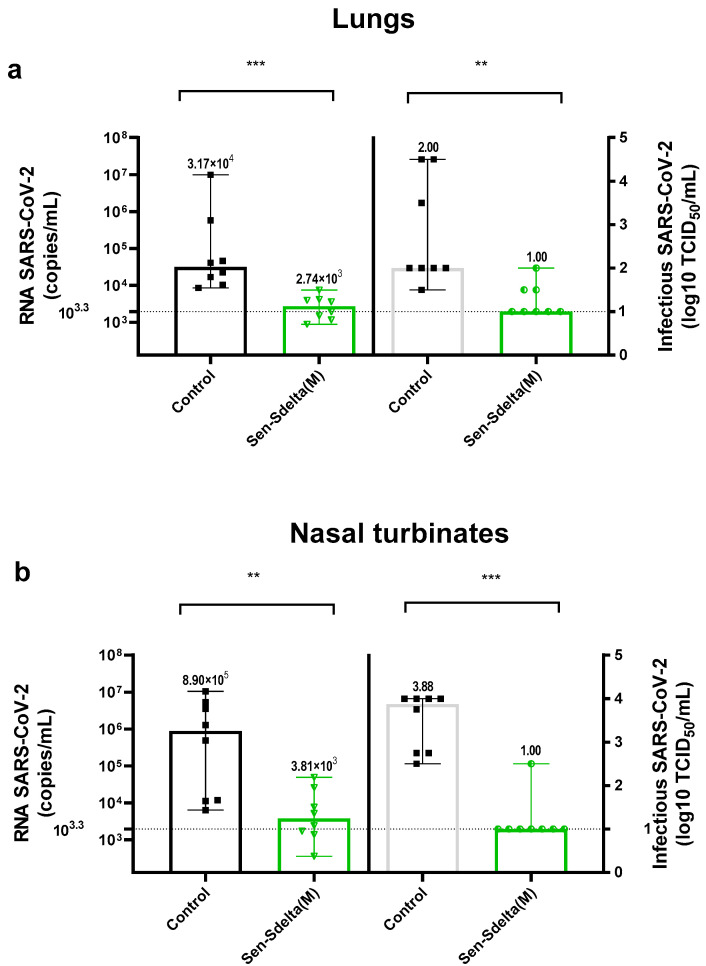
The viral load in the lungs (**a**) and nasal turbinates (**b**) of BALB/c mice measured 5 days after intranasal infection with the SARS-CoV-2 gamma VOC. The viral RNA concentration (copies/mL) in 10% tissue homogenates is indicated on the left side, while the virus infectious titer on Vero E6 cells (log_10_ TCID_50_/mL) is shown on the right side. Each data point represents an individual value, with the tops of the histograms indicating the group medians, and vertical lines representing the 95% confidence intervals (95% CI). Horizontal drop lines denote the thresholds for SARS-CoV-2 RNA (≤10^3.3^ copies/mL) and infectious titers (≤1 log_10_ TCID_50_/mL). ** *p* < 0.01; *** *p* < 0.001.

**Figure 5 vaccines-12-00783-f005:**
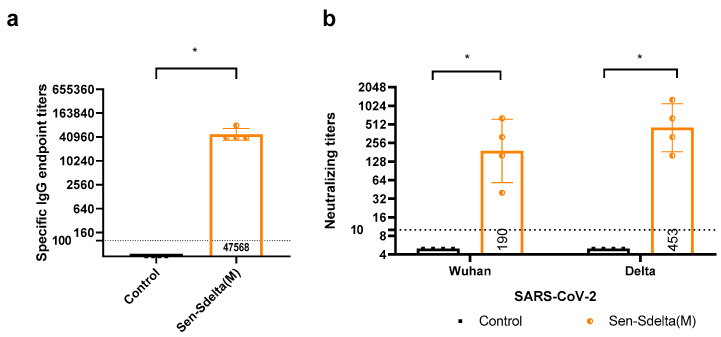
The specific IgG endpoint titers against the delta variant of SARS-CoV-2 measured by ELISA (**a**), and the neutralizing titers against the Wuhan and delta variants determined through neutralization assay on Vero E6 cells (**b**). Each data point represents an individual titer, with the tops of the histograms indicating the GMTs, and vertical lines representing the 95% CI. Horizontal drop lines denote the titers’ thresholds for IgG (<1:100) and neutralizing titers (<1:10), with values below these thresholds estimated as 1:40 and 1:5, respectively, for GMT calculations. * *p* < 0.05.

**Figure 6 vaccines-12-00783-f006:**
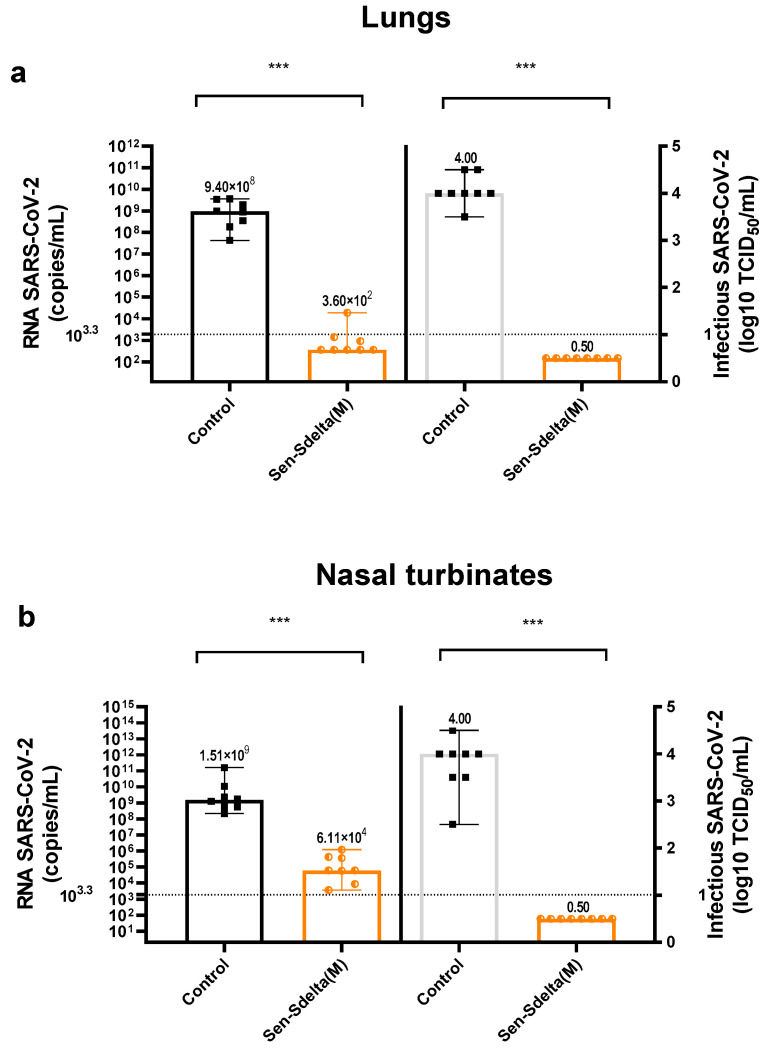
The viral load in the lungs (**a**) and nasal turbinates (**b**) of Syrian hamsters measuring 6 days after intranasal infection with the SARS-CoV-2 delta VOC. Viral RNA concentration (copies/mL) in 10% tissue homogenates is indicated on the left side; virus infectious titer on Vero E6 cells (log_10_ TCID_50_/mL) is presented on the right side. Each data point represents an individual value, with the tops of the histograms indicating the group medians, and vertical lines representing the 95% CI. Horizontal drop lines denote the thresholds for RNA SARS-CoV-2 (≤10^3.3^ copies/mL) and infectious titers (≤log_10_ TCID_50_/mL). *** *p* < 0.001.

**Table 1 vaccines-12-00783-t001:** A set of primers for the amplification of a transgene Sdelta.

Gene	Sequences of Primers (5’…3’)	Restriction Sites
Sdelta	cggaattcgtacgccaccatgtttgtttttcttgttttattgccacta (forward)	BsiWI
	tttatgcatgcgcgctatgtgtaatgtaatttgactcctttgagc (reverse)	BssHII

**Table 2 vaccines-12-00783-t002:** Peptides from the S protein used for stimulation of splenocytes in ELISpot and ICS.

Sequence	MHC Class	Allele
SGTNGTKRF	I	H-2-Dd
YYHKNNKSW	I	H-2-Kd
KYNENGTIT	I	H-2-Kd
VYAWNRKRI	I	H-2-Kd
FERDISTEI	I	H-2-Ld
CGPKKSTNL	I	H-2-Dd
SKPSKRSFI	I	H-2-Dd
KYFKNHTSP	I	H-2-Kd
YPDKVFRSSVLHSTQ	II	H2-IEd
KNIDGYFKIYSKHTP	II	H2-IEd
RFASVYAWNRKRISN	II	H2-IEd, H2-IAd
SNGTHWFVTQRNFYE	II	H2-IEd
YNYKLPDDFTGCVIA	II	H2-IEd
KNKCVNFNFNGLTGT	II	H2-IEd
QPTESIVRF	I	H-2-Kd
VSPTKLNDL	I	H-2-Kd
LLHAPATVCGPKKST	II	H2-IEd
ASVYAWNRKRISN	II	H2-IEd

## Data Availability

The data can be shared upon request.

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
