# Peer review of "Immunogenicity and Protective Efficacy of a Single Intranasal Dose Vectored Vaccine Based on Sendai Virus (Moscow Strain) against SARS-CoV-2 Variant of Concern"

_vaccines, 2024, doi:10.3390/vaccines12070783_

Round 1

Reviewer 1 Report

Comments and Suggestions for Authors

Review Manuscript ID: vaccines-3078724

Immunogenicity and protective efficacy of a single intranasal dose vectored vaccine based on Sendai virus (Moscow strain) against SARS-CoV-2 variant of concern Special Issue: SARS-CoV-2 Variants, Vaccines, and Immune Responses by Kochneva et al is an interesting study reporting on a novel vaccine platform, recombinant Sendai virus. They use this platform to induce protective immunity against various strains of SARS CoV 2 using a mouse and a hamster model of infection. Overall, the manuscript is well written and the in vivo challenge data look promising for the hamster model. However, there are several major weaknesses in the study that contest the conclusions drawn by the authors.

The authors claim 100% clearance of the virus in vaccinated hamsters. However, the detection level was stated to be 10to 3.3 copies/mL and thus, no claim for 100% elimination can be done. This should be rephrased as reduction below detection level.

The authors state that they observed a CD8 immune response. However, in the splenocyte assay peptides were offered, not tetramers- how could be CD8 positive antigen specific cells really be detected? Why were not CD8 cell specific responses attempted to be detected? Furthermore, an important control, offering peptides unrelated to the spike protein is missing. Without that, one cannot really conclude that the observed cytokine response is specific to the spike protein.

The detection of bands in western blot employing low diluted convalescent plasma is not a proof that the protein produced is immunologically identical with the S protein. The authors should have probed the western also with plasma from uninfected individuals or sequenced the presumptive protein.  

Additional weaknesses are listed below.

Introduction:

Authors should address the pathogenicity of Sendai virus in rodents in their introduction.

Material methods:

The authors should provide more detail or include references with detailed methodology for red blood cell preparations, virus extractions- clarifying how cells were lyzed (unless just by one round of freeze-thawing?), hemagglutination assay, western immunoblotting (e.g., what was used for blocking, buffers?)

Did the authors determine isotype subclasses in immune serum of their animals? It would be useful to include that information to better understand the immune response as antibody subtype vary in the effects.

The addition of 50 ug/mL gentamicin when assessing splenocyte immune responses seems high and toxic. Can the authors verify that it was indeed 50 ug/mL and address whether toxic effects had been observed?

For the cytokine assays, only spike protein peptides were used. An important control would have been to include peptides unrelated to the spike protein to conclude that the T cell responses were specific.

How can the authors detect CD8 cell responses by only offering peptide and not tetramers?

Results

Showing the western blot data is insufficient to confirm the presence of the spike protein as western was probed with human convalescence serum at high concentration. The authors could attempt mass spectrum analysis or detection with a specific high tittered antiserum or show in addition a western using normal human plasma at the same concentration (or even better get a paired plasma or serum sample).

The authors cannot conclude 100% protection, they only can state that the viral infectious dose was reduced below detection level.

Fig 1b. There is no need to remove duplicate lanes. The author should show the entire original blot with all lanes.

Discussion

The authors should better discuss why amniotic fluid contained primarily cleaved spike protein and why the immune response and protection varied so much between the mice and hamsters.

The authors used a low dilution (high concentration of plasma in their western blot. This is not proof for the presence of spike protein and this statement should be removed from the discussion.

References

These need to be updated when addressing the reviewers comments (back ground on Sendai virus, methods, difference between mice and hamster responses).

Author Response

Comment 1: The authors claim 100% clearance of the virus in vaccinated hamsters. However, the detection level was stated to be 10to 3.3 copies/mL and thus, no claim for 100% elimination can be done. This should be rephrased as reduction below detection level.

Responce 1: We agree with this comment. Corresponding changes to the text are marked in red and can be found in the abstract on page 1, lines 21-22; on page 13, lines 475-477 and on page 14, lines 533-535.  

 Comment 2: The authors state that they observed a CD8 immune response. However, in the splenocyte assay peptides were offered, not tetramers- how could be CD8 positive antigen specific cells really be detected? Why were not CD8 cell specific responses attempted to be detected? Furthermore, an important control, offering peptides unrelated to the spike protein is missing. Without that, one cannot really conclude that the observed cytokine response is specific to the spike protein.

Responce 2: We were not able to perform detection of virus-specific CD8 cells using MHC tetramers tetramer detection because, unfortunately, at the time of the experiment we could not find commercially available tetramers or pentamers specific to the SARS-COV 2 S protein for BALB/c mice (allele H-2D).  For example, the best-known manufacturer of MHC tetramers ProImmune (https://www.proimmune.com) produced SARS-COV 2 specific tetramers only for humans. The manufacturer NIH Tetramer Core Facility (https://tetramer.yerkes.emory.edu/reagents/sars-cov-2-covid-19) has MHC tetramers for BALB/c mice (allele H-2D) to only one epitope from the N protein. To assess functional antigen-specific T cells, methods such as the intracellular cytokine staining (ICS) assay or the interferon-(IFN-)γ Enzyme-Linked Immuno-Spot Assay (ELISpot) are traditional and widely used. Synthetic peptides represent the most accessible and convenient format for in vitro T-cell stimulation. We believe that a virus-specific response was recorded because splenocytes obtained from a control not immunized group did not respond to the addition of peptides from the SARS-COV-2 S protein with cytokine release.

Comment 3: The detection of bands in western blot employing low diluted convalescent plasma is not a proof that the protein produced is immunologically identical with the S protein. The authors should have probed the western also with plasma from uninfected individuals or sequenced the presumptive protein.  

Responce 3: We agree with this comment. We removed the phrase about the immunologic identity of recombinant and natural S proteins from the Discussion.

Comment 4: Authors should address the pathogenicity of Sendai virus in rodents in their introduction.

Responce 4: We agree with this comment and have inserted the relevant information in the Introduction section. This change is marked in red and can be found on page 2, lines 58-65.

Comment 5: The authors should provide more detail or include references with detailed methodology for red blood cell preparations, virus extractions- clarifying how cells were lyzed (unless just by one round of freeze-thawing?), hemagglutination assay, western immunoblotting (e.g., what was used for blocking, buffers?)

Responce 5: We have detailed descriptions of the hemagglutination and Western blot assay methods, including detailed descriptions of the composition of the lysing and blocking buffers and the erythrocyte preparation buffer. These changes are marked in red and can be found on page 3, lines 126-127; page 4, lines 152-157 and 183-196.

Comment 6: Did the authors determine isotype subclasses in immune serum of their animals? It would be useful to include that information to better understand the immune response as antibody subtype vary in the effects.

Responce 6: No, we did not define subclasses of isotypes. We determined the total amount of IgG class antibodies.

Comment 7: The addition of 50 ug/mL gentamicin when assessing splenocyte immune responses seems high and toxic. Can the authors verify that it was indeed 50 ug/mL and address whether toxic effects had been observed?

Responce 7: 50 µg/mL of gentamicin is the standard concentration of antibiotic and is recommended in many methodological manuals, e.g. Kalyuzhny. Handbook of ELISPOT: Methods and Protocols (Methods in Molecular Biology, Vol. 792, 2011). 2nd Edition. ISBN-10: 9781617793240. 

Comment 8: For the cytokine assays, only spike protein peptides were used. An important control would have been to include peptides unrelated to the spike protein to conclude that the T cell responses were specific.

Responce 8: We believe that a virus-specific response was recorded because splenocytes obtained from a control group of animals not immunized with the vaccine did not respond to the addition of peptides from the SARS-COV-2 S protein with cytokine release.

Comment 9: How can the authors detect CD8 cell responses by only offering peptide and not tetramers?

Responce 9: This comment overlaps with comment 2, to which we have partially responded. ELISpot and ICS allow to detect virus-specific T cell response by stimulating lymphocytes with virus-specific peptides. Synthetic peptides represent the most accessible and convenient format for stimulating T cells in vitro. Since CD8+ and CD4+ T cells recognize short epitopes (classically 8–12 amino acids (aa) on HLA class I and 13–18 aa on HLA class II), “minimal” synthetic short peptides can be used when the epitopes and/or the HLA restriction are known or should be precisely identified.

Comment 10: Showing the western blot data is insufficient to confirm the presence of the spike protein as western was probed with human convalescence serum at high concentration. The authors could attempt mass spectrum analysis or detection with a specific high tittered antiserum or show in addition a western using normal human plasma at the same concentration (or even better get a paired plasma or serum sample).

Responce 10: We still believe that the data presented support the expression of S protein within recombinant Sendai virus, taking into account all available controls in Figure 1b:

  • Reconvalescent serum interacts with the positive control, which is purified S protein in monomeric and trimeric form (lane 8);
  • Reconvalescent serum does not interact with lysates of LLC-MK2 cells uninfected or infected with the original Sendai virus (lanes 1 and 2), but forms clear interaction bands with lysate of Sen-Sdelta(M) infected cells (lane 3);
  • Reconvalescent serum does not interact with the allantois fluid-derived preparation of the original Sendai virus, but forms a clear complementary band with the allantois fluid-derived preparation of recombinant Sen-Sdelta(M) (paired samples);
  • all the protein bands that appeared in the samples with Sen-Sdelta(M) have molecular masses that are consistent with the known mass of the S protein and its S1/S2 fragments, as well as with the trimeric and monomeric forms of the S protein in the control sample (lane 8).

Comment 11: The authors cannot conclude 100% protection, they only can state that the viral infectious dose was reduced below detection level.

Responce 11: This comment is part of comment 1, to which we responded. We have adjusted everything in accordance with the reviewer's comment.

Comment 12: Fig 1b. There is no need to remove duplicate lanes. The author should show the entire original blot with all lanes.

Responce 12: The lanes that we have removed in the final Figure 1b do not carry any additional meaning and represent slightly reduced volumes of the same samples loading. Such an adjustment is allowed by journal rules, and we took advantage of it to avoid overloading the figure.

Comment 13: The authors should better discuss why amniotic fluid contained primarily cleaved spike protein and why the immune response and protection varied so much between the mice and hamsters.

Responce 13: We have added two additional sections to the Discussion, one discussing the reviewer's question "Why allantoic fluid contained primarily cleaved spike protein" and the other discussing "Why the immune response and protection varied so much between the mice and hamsters". Additional references are included in both sections. These changes are marked in red and can be found on page 14, lines 498-500 and page 15, lines 540-556.

Comment 14: The authors used a low dilution (high concentration of plasma in their western blot. This is not proof for the presence of spike protein and this statement should be removed from the discussion.

Responce 14: This comment overlaps with comment 10, in response to which we sought to substantiate the validity of our conclusions.

Comment 15: References. These need to be updated when addressing the reviewer’s comments (back ground on Sendai virus, methods, difference between mice and hamster responses).

Responce 15: The references have been updated to reflect the text corrections made.

Reviewer 2 Report

Comments and Suggestions for Authors The paper addresses whether Sendai virus can be used as an effective vehicle for SARS-CoV-2 vaccination in vivo. The approach is novel, though, as the authors acknowledge this is not necessarily needed at this time. It is, however, is a good "proof of concept" that these vaccines might work in the future.   This is a paper that "simply" shows that the vaccine works, so in that context fulfils the aims really well. And all conclusions are valid based on data presented.   I have a small concern about the fact that "negative" data is presented below the detection limit. Can the authors confirm that "negative" data was treated as at the detection limit for statistical purposes. As these are at the limit, though, I am not sure the stats are the critical point (i.e. The authors can simply claim that the virus is reduced to the detection limit.

Comments on the Quality of English Language

No issues of note.

Author Response

Comment 1: I have a small concern about the fact that "negative" data is presented below the detection limit. Can the authors confirm that "negative" data was treated as at the detection limit for statistical purposes. As these are at the limit, though, I am not sure the stats are the critical point (i.e. The authors can simply claim that the virus is reduced to the detection limit.

Responce 1:Yes, that's what we wrote in the description of the results: samples (tissue homogenates or serum) did not contain detectable levels of infectious SARS-CoV-2 or vRNA or antibodies. Throughout the text, these places are marked in red.

But to calculate GMT we used some recalculations of negative values accepted in statistical processing.To calculate the antibody titer in a neutralization reaction, all samples with titers below the threshold are usually assigned a titer equal to half the threshold, i.e. 1:5 in our case. For calculation of antibody titer in ELISA, a similar approach may be used or recalculation is made to the maximum value of optical density in samples with titers below the threshold value (ODmax × 100), resulting in a value of 1:40 in our case (https://doi.org/10.1038/s41598-022-10281-1; https://doi.org/10.3390/v13060996; https://doi.org/10.3389/fimmu.2022.910136; https://doi.org/10.1038/s41541-022-00551-4 and others). The same approach is used for the statistical treatment of negative values of infectious virus titers in tissue or cell homogenate samples, namely, all samples with titers below the threshold are usually assigned a titer equal to half the threshold, as we did (https://doi.org/10.3389/fmicb.2022.967019; https://doi.org/10.1038/s41541-023-00699-7 and others).

Reviewer 3 Report

Comments and Suggestions for Authors

(1) Mice are susceptible to Sendai virus, so it is recommended to detect the pre-existing antibodies against Sendai virus in mice; (2) Modify all figures, format is should be unified, unified graph fonts, each group use with the same style and color; (3) Pay attention to modify the chapter format, such as 3.1, 3.2 font inconsistent.

Author Response

Comment 1: Mice are susceptible to Sendai virus, so it is recommended to detect the preexisting antibodies against Sendai virus in mice.

Responce 1: We did not pre-test mice for the presence of antibodies to Sendai virus. All animals were obtained from the nursery of laboratory animals of the FBRI SRC VB “Vector”, where animals are kept under aseptic conditions and are routinely randomly tested for specific pathogens by ELISA and PCR. For mice, testing includes pathogens such as ectromelia (mouse pox) and Sendai virus.

Comment 2: Modify all figures, format is should be unified, unified graph fonts, each group use with the same style and color.

Responce 2: We have corrected the figures, especially figure 3.

Comment 3: Pay attention to modify the chapter format, such as 3.1, 3.2 font inconsistent.

Responce 3: We've corrected the chapter format.

Round 2

Reviewer 2 Report

Comments and Suggestions for Authors

Thank you for the clarification